# Machine Learning Parameterization of the Multi-scale Kain-Fritsch (MSKF) Convection Scheme and stable simulation coupled in WRF using WRF-ML v1.0

Xiaohui Zhong[1,*], Xing Yu[2], and Hao Li[1,*]

[1]Fudan University, Shanghai, 200433, China
[2]Shenzhen Institute of Artificial Intelligence and Robotics for Society, Guangdong, 518000, China
[*]These authors contributed equally to this work.

**Correspondence:** Xing Yu (yuxing@cuhk.edu.cn)

**Abstract.** Warm-sector heavy rainfall along the South China coast poses significant forecasting challenges due to its localized nature and prolonged duration. To improve the prediction of such high-impact weather events, high-resolution numerical weather prediction (NWP) models are increasingly used to more accurately represent topographic effects. However, as these models' grid spacing approaches the scale of convective processes, they enter a "gray zone" where the models struggle to fully resolve the turbulent eddies within the atmospheric boundary layer, necessitating partial parameterization. The appropriateness of applying convection parameterization (CP) schemes within this gray zone remains controversial. To address this, scale-aware CP schemes have been developed to improve the representation of convective transport. Among these, the multi-scale Kain-Fritsch (MSKF) scheme enhances the traditional Kain-Fritsch (KF) scheme, incorporating modifications that facilitate its effective application at spatial resolutions as high as 2 km. In recent years, there has been an increasing application of machine learning (ML) models across various domains of atmospheric sciences, including efforts to replace conventional physical parameterizations with ML models. This work introduces a multi-output bidirectional long short-term memory (Bi-LSTM) model intended to replace the scale-aware MSKF CP scheme. This multi-output Bi-LSTM model is capable of simultaneously predicting the convection trigger while also modeling the associated convective tendencies and precipitation rates with high performance. Data for training and testing the model are generated using the Weather Research and Forecast (WRF) model over South China at a horizontal resolution of 5 km. Furthermore, this work evaluates the performance of the WRF model coupled with the ML-based CP scheme against simulations with traditional MSKF scheme. The results demonstrate that the Bi-LSTM model can achieve high accuracy, indicating the promising potential of ML models to substitute the MSKF scheme in the gray zone.

## 1 Introduction

Warm-sector heavy rainfall often occurs in South China during the pre-flood season, primarily influenced by the East Asian summer monsoon (Ding, 2004). These rainfall events are characterized by intense and localized precipitation over limited area. Despite their small scale, such unexpected and extreme warm-sector rainfall can cause significant damage, including

flooding homes and vehicles, destroying crop fields, and endangering lives, leading to economic losses ranging from millions
to even billions of dollars (Tao, 1981; Zhao et al., 2007; Zhong et al., 2015). Accurately predicting warm-sector heavy rainfall

with Numerical Weather Prediction (NWP) models is challenging due to the complex interaction of various factors, such as
the low-level jet (LLJ), land-sea contrast, topography, and urban landscape (Zhong and Chen, 2017; Luo et al., 2017; Jian
et al., 2002; Di et al., 2006; Xia and Zhao, 2009; Zhang and Ni, 2009). The complex terrain and heterogeneous land surface
of South China region are crucial in promoting active convection. Previous studies (Giorgi et al., 2016; Mishra et al., 2018;
Schumacher et al., 2020; Onishi et al., 2023) have demonstrated that higher spatial resolution improves the performance of

30 convective rainfall forecasts by more accurately resolving topographic features. Acknowledging the importance of resolution
in forecasting severe convective weather, both the Chinese government and the community have increasingly supported the
development of high-resolution operational forecast models specifically designed for warm-sector rainstorms and sudden local
rainstorms. In early 2017, the China Meteorological Administration (CMA) launched an initiative to develop a comprehensive
framework for evaluating the forecast performance of all available models, including high-resolution regional models, and

35 advancing key technologies for forecasting high-impact weather.

The increased computational resources has facilitated a shift towards the implementation of regional NWP models with
increasingly finer grid spacings, typically within the range of 1 to 10 km. However, when the model grid spacing approaches
the scale of convection, entering the so-called "gray zone" (Wyngaard, 2004; Hong and Dudhia, 2012), cumulus convection
transitions from being completely unresolved to partially resolved. Theoretically, the accurate representation of the smallest

turbulent scales, achievable only through Direct Numerical Simulation (DNS) at resolutions from millimeters to centimeters
(Jeworrek et al., 2019), still requires the use of parameterization of turbulence or convection in weather modeling. There is
ongoing debate regarding the efficacy of employing convection parameterization (CP) within the gray zone. Several studies
(Chan et al., 2013; Johnson et al., 2013) have found that reducing horizontal grid spacing to below 4 km while using CP scheme,
does not enhance and may even degrade, precipitation forecast performance. In contrast, other studies (Lean et al., 2008;

Roberts and Lean, 2008; Clark et al., 2012) showed that forecasts with a horizontal grid spacing of 1 km, where convection is
explicitly resolved, yielded more accurate spatial representation of accumulated rainfall over 48 hours compared to forecasts
using 12 km and 4 km grid spacings. This discrepancy in research findings, with some indicating no benefit from finer grid
spacing and others suggesting improved forecast accuracy, seems to stem from the application of the CP at scales beyond its
originally intended operational range. Therefore, it remains unclear if utilizing any CP schemes in the gray zone is effective

for predicting localized warm-sector heavy rainfall.

To enhance prediction accuracy in the gray zone, researchers have developed scale-aware CP schemes. These schemes
dynamically parameterize convective processes based on the horizontal grid spacing, thus facilitating seamless transitions be-
tween different spatial scales. A pivotal study by Jeworrek et al. (2019) demonstrated that two specific scale-aware CP schemes,
Grell-Freitas (Grell and Freitas, 2014) and multi-scale Kain-Fritsch (MSKF) (Zheng et al., 2016), surpassed conventional CP

schemes in predicting both the timing and intensity of precipitation over the Southern Great Plains of the United States. Ad-
ditionally, Ou et al. (2020) showed that the MSKF scheme outperformed other CP schemes, including Grell-3D Ensemble
(Grell and Dévényi, 2002) and New Simplified Arakawa-Schubert (Han and Pan, 2011), in precipitation simulation. This was

evidenced by its lower root mean squared error (RMSE) values when compared against in-situ observations and satellite data. Despite the increasing adoption of these scale-aware schemes due to their superior performance, it is crucial to acknowledge that their efficacy also rely on various empirical parameters (Villalba-Pradas and Tapiador, 2022). Therefore, developing specialized CP schemes for the gray zone in NWP models continues to be a significant challenge.

In recent years, an increasing number of studies have investigated the use of machine learning (ML) models as alternatives to conventional physics-based CP schemes. These ML-based schemes have demonstrated potential for efficacy across various horizontal resolutions, benefiting from being trained on data from simulations that operate at varying grid resolutions (Yuval and O'Gorman, 2020). Unlike conventional CP schemes, which often rely on assumptions such as convective quasi-equilibrium (Arakawa, 2004), ML-based parameterization schemes do not require such assumptions. Notably, random forests (RFs) and fully-connected (FC) neural networks (NNs) have become predominant ML models for CP schemes in previous studies. RFs offer the advantages of inherently enforcing physical constraints, such as energy conservation and non-negative surface precipitation, essential for maintaining stable simulations. O'Gorman and Dwyer (2018) demonstrated RFs' capability to emulate moist convection in an aquaplanet general circulation model (GCM), maintaining stability and effectively reproducing key climate statistics. Furthermore, Yuval and O'Gorman (2020) employed the coarse-grained output from a high-resolution three-dimensional (3D) GCM model, simulated on an idealized equatorial beta plane, to train the RF parameterization. They showed that the RF parameterization is capable of reproducing the climate of the high-resolution simulation at coarser resolutions. However, FC NNs offer several advantages over RFs, such as the potential for higher accuracy and lower memory requirements. Krasnopolsky et al. (2013) introduced a stochastic CP scheme using an ensemble of 3-layer NNs, trained with data generated by a cloud-resolving model (CRM) during the TOGA-COARE [1] experiment, demonstrating its capacity for generating reasonable decadal climate simulations across a broader tropical Pacific region when incorporated into the National Center of Atmospheric Research (NCAR) Community Atmospheric Model (CAM). Similarly, Gentine et al. (2018) leveraged deep NN (DNN) trained on data from idealized and aquaplanet simulations performed using the SuperParameterized Community Atmosphere Model (SPCAM). The DNN predicts temperature and moisture tendencies due to convection and clouds, as well as the cloud liquid and ice water contents. Additionally, Rasp et al. (2018) successfully implemented an NN-based parameterization in a global GCM on an aquaplanet, conducting stable prognostic simulations over multiple years that accurately reproduced the climatology of SPCAM and capturing crucial aspects of variability, including extreme precipitation and realistic tropical waves. However, Rasp (2020) also found that minor changes to the configuration rapidly led to simulation instabilities, underscoring the need to address the robustness of NN parameterizations in GCMs. Yuval et al. (2021) developed a FC NN to that predicts subgrid fluxes instead of tendencies, incorporating physical constraints from coarse-grained high-resolution atmospheric simulation in an idealized domain. Brenowitz and Bretherton (2018, 2019) proposed a novel loss function designed to minimize accumulated prediction error over multiple time steps to enhance long-term stability and accuracy, by excluding upper atmospheric humidity and temperature from the input. Nonetheless, the approach of removing certain variables from the

---

[1]TOGA-COARE is an acronym for Tropical Ocean Global Atmospheres/Coupled Ocean Atmosphere Response Experiment. It is an international research program that investigates the interaction or coupling of the ocean and atmosphere in the western Pacific warm pool region from November 1992 to February 1993, encompassing 120 days of field experiments involving the deployment of oceanographic ships, moorings, drifters, and Doppler radars (ship, land, air).

input is relatively rudimentary, demanding additional research to enhance the stability of NN-based parameterizations when integrated into the model.

Previous studies have predominantly used FC NNs to emulate convection, while more advanced NN structures have the potential to achieve higher accuracy. In a pioneering study, Han et al. (2020) explored the use of a deep residual convolutional NN (ResNet) (He et al., 2016) for the emulation of convection and cloud parameterization in the SPCAM model using a realistic configuration. They compared the performance of ResNet with various NN architectures, including a FC DNN, a DNN with skip connections, and a convolutional NN (CNN) without skip connections. The results revealed that ResNet and CNNs without skip connections outperformed FC NNs and DNNs with skip connections in accuracy, with ResNet and CNNs without skip connections showing comparable performance. This finding highlights the significant role of convolutions in enhancing accuracy. Furthermore, Yao et al. (2023) evaluated multiple ML model structures for simulating atmospheric radiative transfer processes, encompassing FC NNs, CNNs, bidirectional recurrent-based NNs (RNNs), transformer-based NNs (Vaswani et al., 2017), and Fourier Neural Operators (FNO (Li et al., 2020)). Their results indicated that models capable of preceiving global context of the entire atmospheric column significantly outperformed FC NNs and CNNs. Particularly, the bidirectional long short-term memory (Bi-LSTM) achieved the highest levels of accuracy. Similar to radiative transfer modeling, Han et al. (2020) also emphasized the importance of ML having a global perspective of the entire atmospheric column for ML models in convection modeling. They demonstrated that increasing the depths of CNNs from 4 to 22 layers significantly improved model accuracy, a benefit primarily attributed to the expansion of the receptive field in deeper CNN layers. Therefore, ML models that integrate both global and local perception capabilities are better suited for developing ML-based CP schemes.

Previous research have mostly focused on replacing CP schemes in GCM models with ML models for climate forecasting. The complexity of CP schemes in weather forecasting models surpasses that in GCMs (Arakawa, 2004). Generally, CP schemes in GCMs, whether in explicit or implicit form, assume that both the horizontal grid size and the temporal intervals for physics implementation are significantly larger and longer compared to the grid size and duration of individual moist-convective elements. In contrast, CP schemes in high-resolution models must account for dependencies on both the model's resolution and the time interval for implementing the physics (Arakawa, 2004). The ultimate goal is to develop ML models, based on data from super-parameterization or cloud-resolving models, to replace conventional CP schemes in weather fore-casting models. This replacement seeks to reduce uncertainties and improve the efficacy of ML parameterizations. This study represents an initial effort to employ a ML model as an alternative to conventional CP schemes in weather forecasting models. For our dataset, we used the Weather and Research Forecasting (WRF) (Skamarock et al., 2021) model that covers the South China region, incorporating the scale-aware MSKF scheme employed as the CP scheme. The MSKF scheme, an improved version of the Kain-Fritsch (KF) scheme (Kain and Fritsch, 1990, 1993; Kain, 2004), aims to mitigate the overestimation of precipitation, address the premature convection trigger issue, particularly evident in overestimating precipitation during sum-mer. To address these issues, the MSKF incorporates a scale-dependent capability, such as modifying the formulation of the convective adjustment timescale. This vital parameter, which determines the intensity and duration of convection, has been made dynamic and dependent on grid resolution (Zhang et al., 2021b). Furthermore, we utilize a Bi-LSTM model to emulate

the convective processes and couple it with the WRF model through the WRF-ML coupler developed by Zhong et al. (2023a). The performance of the ML-based CP scheme is evaluated in both offline and online settings.

The paper is structured as follows. Section 2 provides a description of the WRF model for data generation, as well as the input and output data of the ML model. In Section 3, the original and the ML-based MSKF scheme is introduced. The results for both offline and online testing of the ML-based MSKF scheme are presented in Section 4. Finally, Section 5 presents the summary and conclusion.

## 2 Data

### 2.1 Data generation

The dataset was generated by running the WRF model version 4.3 (Skamarock et al., 2019, 2021). The following subsections provide a comprehensive explanation of the WRF model configurations, as well as the input and output variables employed in the development of the ML-based CP scheme.

The WRF model is compiled using the GNU Fortran (gfortran version 7.5.0) compiler with the "dmpar" option. The WRF model is run using the domain configuration illustrated in Figure 1. The WRF model is configured with a single domain consisting of 44000 grid points, with a horizontal grid spacing of 5 km and dimensions of $220 \times 200$ grid points in the west-east and north-south directions. The model consists of 45 vertical levels (i.e., 44 vertical layers), with a model top at 50 hPa. Additionally, the WRF model is configured with physics schemes, including WSM 6-class graupel scheme (Hong and Lim, 2006) for microphysics, Bougeault-Lacarrère (BouLac) scheme (Bougeault and Lacarrère, 1989) for planetary boundary layer (PBL) mixing, the Monin-Obukhov (Janjic) surface layer scheme (Janjic, 1996), the Unified Noah model (Livneh et al., 2011) for land surface, RRTMG for both shortwave and longwave radiation (Iacono et al., 2008), and MSKF (Zheng et al., 2016) for cumulus. The time step used for all WRF simulations is set to 15 seconds.

The initial and boundary conditions for this work were derived from the ERA5 reanalysis dataset, which was provided by the European Centre for Medium-range Weather Forecast (ECMWF) (Hersbach et al., 2020). The ERA5 reanalysis dataset used in this study has a horizontal resolution of $0.25°$ and consists of 29 pressure levels below 50 hPa. To create a dataset for developing the ML model, the WRF simulations were initialized at 12 UTC and conducted 9 times every 2 days, specifically from May 20th, 2022 to June 5th, 2022. Throughout the simulations, the MSKF scheme was called every 5 model minutes, generating outputs at each call. The simulations ran for 36 hours each time, with the first 24 hours used for training and the last 12 hours for validation. Therefore, the total number of training samples is 114,444,000 [2] while the offline validation set contains 57,024,000 [3] samples.

Furthermore, given the possible discrepancy between offline performance, we conducted experiments that coupled the ML-based MSKF scheme with WRF model. This coupling aims at evaluating the online efficacy of the ML-based MSKF scheme by

---

[2] 114,444,000 = 44000 × 9 × (24 × 60 / 5 + 1)

[3] 57,024,000 = 44000 × 9 × 12 × 60 / 5

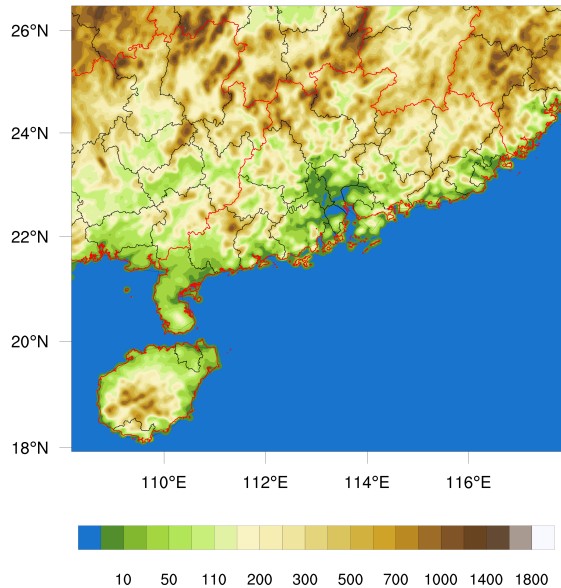

**Figure 1.** Digital evaluation data of the single WRF domain with horizontal resolution at $5^{\circ}$. Red lines are the province borderlines, and black lines are the city borderlines.

comparing it with the original WRF simulations. These simulations were performed 4 times every 2 days, with each simulation
extending over a period of 168 hours (7 days). The initialization days spanned from June 12th, 2022 to June 18th, 2022.

## 2.2    Input and output data

Table 1 presents a comprehensive list of the input and output variables used in this study, consistent with those utilized in the original MSKF scheme. There are 17 variables exclusively used as input, while 9 variables serve as both input and output. Specifically, the output variable "raincv", representing the time-step precipitation due to convection, is calculated through
multiplying the precipitation rate by the model's time step. Among all the variables, 5 are two-dimensional (2D) surface variables, while the remaining ones are 3D variables characterized by 44-layer vertical profiles. Moreover, the ML model used in this study incorporates 4 derived variables as input. These variables consist of a 2D Boolean variable indicating convection triggering based on "nca" values, the pressure difference across adjacent vertical levels, the saturated water vapor mixing ratio, and relative humidity. Furthermore, the output "w0avg", which depends on vertical wind component (w) and input "w0avg", is
also included as an input to model. In total, the ML model utilizes 27 input variables.

     The variable "nca" represents the cloud relaxation time and must be an integer multiple of the model time step. For all WRF model simulations conducted in this study, a fixed time step of 15 seconds is used. Thus, "nca" is expected to be exactly divisible by 15. To eliminate dependence on the specific model time step, "nca" is divided by the model time step before

normalization is applied during model training. Moreover, within the MSKF scheme, "nca" plays a crucial role in determining the triggering of convection. Convection is triggered when "nca" is equal to or exceeds half of the model time step.

To ensure consistency with the dimensions of the 3D variables, the surface variables are padded by duplicating the values of the surface layer for all layers before feeding them into the model. Prior to utilizing the variables in the Bi-LSTM model for training and validation, normalization is applied to ensure uniformity in the magnitudes of all the variables. Each variable is divided by the maximum absolute value in the atmospheric column (for 3D variables) or at the surface (for surface variables).

**Table 1.** Definition of all the input and output variables, and whether they are surface or 3D variables and their corresponding units. There are 44 model layers.

| Type | Variable name | Definition | type | Unit |
|---|---|---|---|---|
| Input | u | meridional wind component | 3D | m/s |
| | v | zonal wind component | 3D | m/s |
| | w | vertical wind component | 3D | m/s |
| | t | temperature | 3D | K |
| | qv | water vapor mixing ratio | 3D | kg/kg |
| | p | pressure | 3D | Pa |
| | th | potential temperature | 3D | K |
| | dz8w | layer thickness | 3D | m |
| | rho | air density | 3D | kg/m$^3$ |
| | pi | Exner function, which is dimensionless pressure and can be defined as: $\left(\frac{p}{p_0}\right)^{R_d/c_p}$ | | |
| | hfx | upward heat flux at surface | surface | W/m$^2$ |
| | ust | u$*$ in similarity theory | surface | W/m$^2$ |
| | pblh | planetary boundary layer height | surface | m |
| Derived Input | p$_{diff}$ | pressure difference between adjacent levels | 3D | Pa |
| | qv$_{sat}$ | saturated water vapor mixing ratio | 3D | kg/kg |
| | rh | relative humidity | 3D | - |
| | trigger | boolean trigger indicating convection triggering | surface | - |
| Input and Output | rthcuten | potential temperature tendency due to cumulus parameterization | 3D | K/s |
| | rqvcuten | water vapor mixing ratio tendency due to cumulus parameterization | 3D | kg/kg/s |
| | rqccuten | cloud water mixing ratio tendency due to cumulus parameterization | 3D | kg/kg/s |
| | rqrcuten | rain water mixing ratio tendency due to cumulus parameterization | 3D | kg/kg/s |
| | rqicuten | cloud ice mixing ratio tendency due to cumulus parameterization | 3D | kg/kg/s |
| | rqscuten | snow mixing ratio tendency due to cumulus parameterization | 3D | kg/kg/s |
| | w0avg | average vertical velocity | 3D | m/s |
| | nca | counter of the cloud relaxation time | 3D | s |
| | pratec | precipitation rate due to cumulus parameterization | surface | mm/s |
| Output | raincv | precipitation due to cumulus paramterization | surface | mm |

 **3   Method**

This section describes the flow chart of original MSKF scheme for determining convection trigger, ML model structures and training, and the evaluation methods.

**3.1   Description of original MSKF module**

The MSKF scheme is a scale-aware adaptation of the KF CP scheme, initially developed by Kain and Fritsch (1990, 1993) and further refined by Kain (2004). Figure 2 illustrates the convection trigger process within the MSKF scheme. At the beginning of each simulation step, the scheme evaluates the variable "nca" to ascertain whether it equals or surpasses a threshold, defined as half the model's time step (dt). Should "nca" equal or exceed the half of dt, there is no need to update convective tendencies or precipitation rates due to ongoing convection. In contrast, a "nca" value below this threshold triggers the MSKF scheme to employ a one-dimensional cloud model. This model calculates a set of variables related to cloud characteristics to evaluate the potential of convection triggering. Essential variables include the lifting condensation level (LCL), convective available potential energy (CAPE), cloud top and base heights, and entrainment rates. The LCL is crucial for determining the emergence of potential convective activities, with a lower LCL favoring more intense convection. CAPE quantifies the buoyant energy available to an air parcel for the formation of deep convective clouds upon reaching its Level of Free Convection (LFC) above the LCL, with higher CAPE values signifying a greater potential for intense convection. The cloud base is generally at the LCL, whereas the cloud top is defined at the altitude where buoyancy becomes negligible. Meanwhile, the vertical extent between the cloud base and top affect the cloud's growth and precipitation potential. The MSKF scheme requires surpassing a specific CAPE threshold to trigger convection. Furthermore, it assesses entrainment rates to measure the impact of ambient air on the evolution of convective system. At grid points where convection is triggered, the MSKF scheme calculates both convective tendencies and precipitation rates; otherwise these values are set to zero. However, the variable "w0avg" is consistently updated, regardless of convection status. Active convection leads to a decrement in "nca" by one model time step for each iteration within WRF model cycle.

**3.2   Description of ML-based MSKF scheme**

In the original MSKF scheme, atmospheric columns are processed sequentially, one at a time, until all horizontal grid points within the domain have been processed. In contrast, the ML-based MSKF scheme processes data in batches, as indicated by "B" in Figure 3, consisting of 27 features across 44 vertical layers. As a result, the input data has dimensions of B $\times$ 27 $\times$ 44. Before being fed into the ML model, the input data undergoes pre-processing through a module incorporating a 1-dimensional (1D) convolutional layer. This module expands the feature dimension from 27 to 64. The following sections provide a comprehensive description of the structures of the ML model.

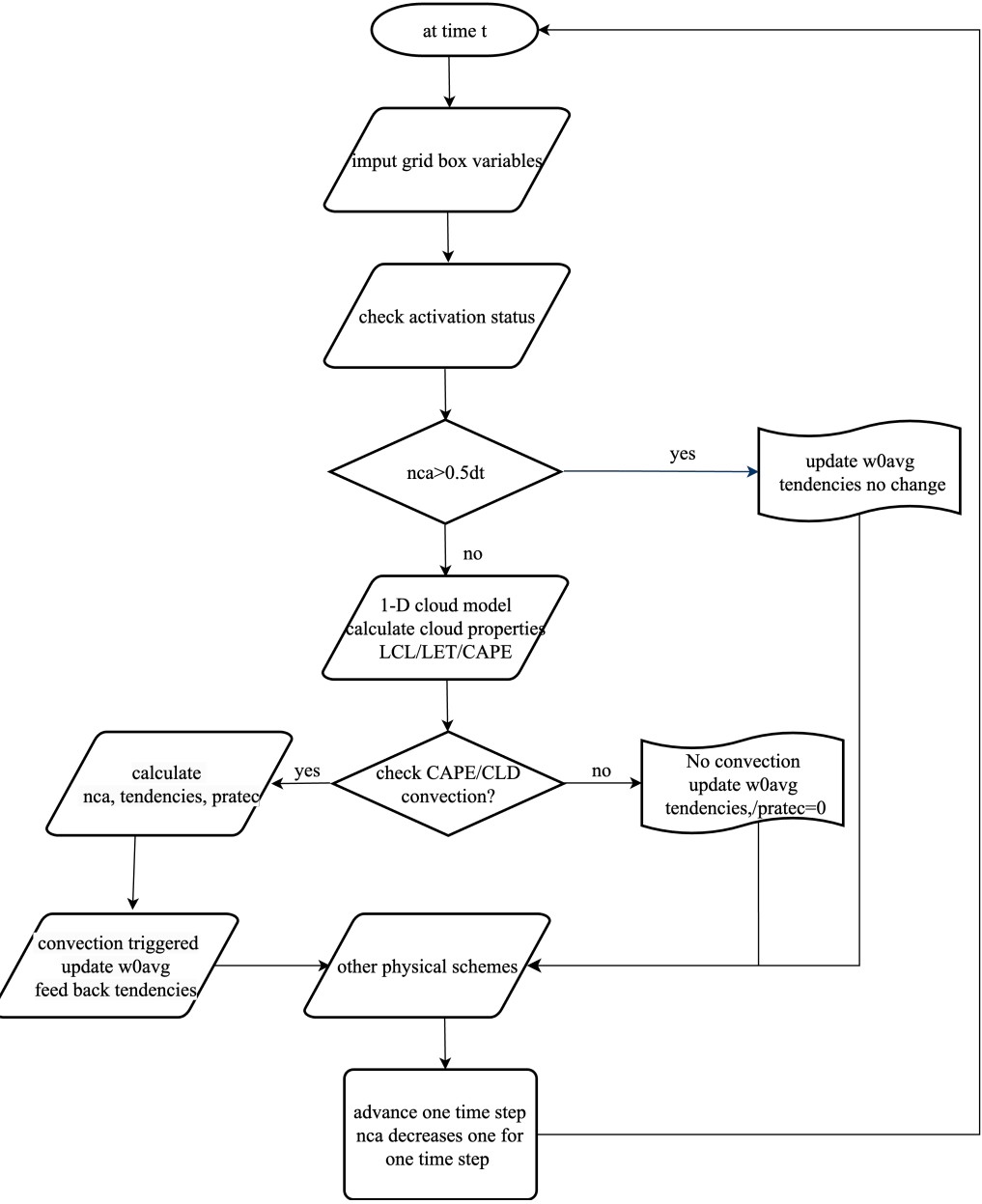

**Figure 2.** A flow chart outlining convection trigger process in the original MSKF scheme.

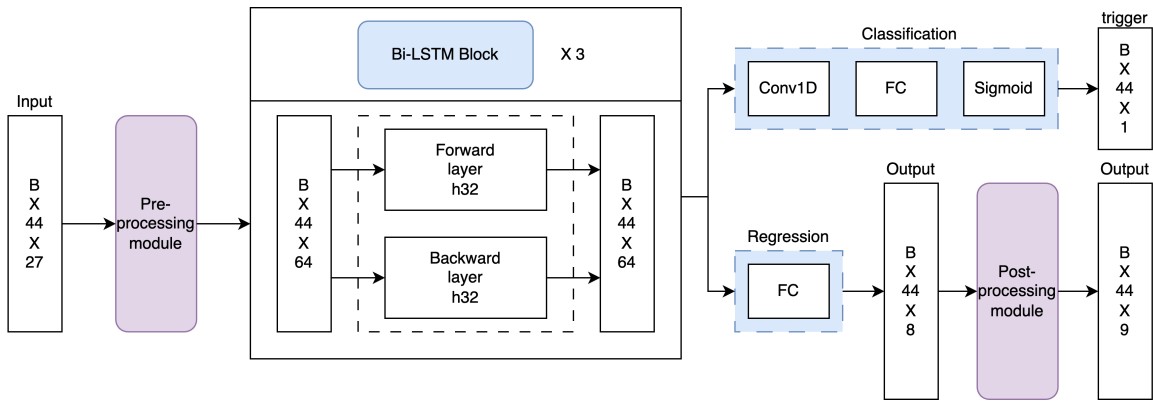

**Figure 3.** The architecture of the multi-output Bi-LSTM model for combined classification and regression predictions.

### 3.2.1 ML model structure

Predicting whether convection is triggered as well as modeling convective tendencies and precipitation rates are two primary objectives of conventional CP schemes. Previous studies have applied ML models to address these objectives, with some dedicated solely to the classification task of convection trigger (Zhang et al., 2021a), while others have independently pursued the regression of convective tendencies (Rasp et al., 2018; Brenowitz and Bretherton, 2019; Wang et al., 2022). However, regression-based models alone may result in inconsistent convective tendencies, leading to conflicting signals for triggering convection at specific grid points (see Figures A3 and A4 in Appendix A). In contrast, models that focus exclusively on classification lack the capability to generate essential tendencies for an effective CP scheme. Therefore, the development of a ML-based CP scheme necessitates the integration of both a binary classification model for the prediction of convection trigger and a regression model for convective tendencies. To address this, we propose a multi-output Bi-LSTM model capable of concurrently conducting regression and classification predictions (Figure 3). Our proposed model consists of a shared Bi-LSTM layer for learning features, a classification subnetwork, and a regression subnetwork. The shared Bi-LSTM layer includes three repeated Bi-LSTM blocks, with each block containing a forward and a backward layer that have a feature dimension of 32. The classification subnetwork is composed of a $1 \times 1$ 1D convolutional layer, a FC layer, and a Sigmoid activation layer. The output of the Sigmoid layer represents the probability distribution of the convection trigger. The binary cross-entropy loss function is employed as the cost function for this classification task. Meanwhile, the regression subnetwork incorporates a FC layer to output precipitation rate, "nca", and convective tendencies. Finally, outputs from both subnetworks are the processed through a post-processing module to ensure their physical consistency (see Figures A5 and A6 in Appendix A), with further details provided in the subsequent subsection.

### 3.2.2 Post-processing module

The post-processing module is designed to ensure physical consistency of all variables. To achieve this, the following rules are applied: 1) At grid points where the input "nca" is equal to or greater than half the value of dt, all other variables remain unchanged as they are still within the convection lifetime. 2) The output "nca" must be an integer. 3) At grid points where convection is predicted to be inactive, all corresponding output variables are default to zero. In addition, the calculation of time-step convective precipitation (raincv) follows the methodology outlined in the previous section 2.2.

### 3.2.3 Model training

As our model incorporates both classification and regression tasks, we optimize its performance by minimizing a multi-task loss function (Ren et al., 2016). The loss function is defined as the sum of the binary cross entropy loss for the convection trigger and a weighted combination of the $L1$ loss for all output variables from the regression subnetwork. The specific formulation of the loss function is as follows:

$$L = \frac{1}{N_{cls}} \sum_{i,j} L_{cls}(p_{i,j}, p_{i,j}^{gt}) + \sum_{c} \lambda_c \frac{1}{N_{reg}} \sum_{i,j} p_{i,j}^{gt} L1_c \tag{1}$$

Here, $i$ and $j$ denote the grid points in the domain. $p_{i,j}$ represents the probability of convection being triggered. The ground-truth label $p_{i,j}^{gt}$ takes a value of 1 if convection is triggered and 0 otherwise. The classification loss $L_{cls}$ is calculated using the binary cross entropy loss. For the regression loss of different variables $c$, $\lambda_c$ functions as a weight that balances the output variables by considering their respective magnitudes. The term $p_{i,j}^{gt} L1_c$ indicates that the $L1$ regression loss is activated only for triggered grid points ($p_{i,j}^{gt} = 1$) and is disabled otherwise ($p_{i,j}^{gt} = 0$). Both loss terms are normalized by $N_{cls}$ and $N_{reg}$, which correspond to the total number of grid points and the number of triggered grid points, respectively.

Adam optimizer (Kingma and Ba, 2014) is used with an initial learning rate of 0.003 update the parameters of the model. Furthermore, the plateau scheduler is implemented to decrease the learning rate by a factor of 0.5 when the loss fails to decrease for five epochs. The model is trained for 150 epochs using a batch size of 44000.

### 3.3 Evaluation methods

The ML-based MSKF scheme is evaluated in both offline and online settings. The offline performance of the ML-based MSKF scheme is evaluated by comparing it against the outputs of the original MSKF scheme using validation dataset, including rthcuten, rqvcuten, rqccuten, rqrcuten, nca, and pratec. The overall model performance metrics include RMSE and correlation coefficient. The mean absolute error (MAE) and mean bias error (MBE) per vertical layer were were calculated using the equation below:

$$MAE_l = \frac{1}{N} \sum_{i=1}^{N} |Y_{ML}(i,l) - Y(i,l)| \tag{2}$$

$$MBE_l = \frac{1}{N} \sum_{i=1}^{N} Y_{ML}(i,l) - Y(i,l) \tag{3}$$

where $Y(i,l)$ and $Y_{ML}(i,l)$ represent the outputs from the original MSKF scheme and ML-based MSKF scheme, respectively. Here, $i$ denotes the horizontal grid point of a vertical profile, $N$ is the number of the horizontal grid points in the domain, $l$ represents the vertical layer index.

## 4   Results

### 4.1   Offline validation of the ML-based MSKF scheme

The offline validation was conducted using data that was not used during the training process. Figure 4 compares the cloud relaxation time (nca), precipitation rate (pratec), and convective tendencies predicted by both the original MSKF scheme and the ML-based MSKF scheme, respectively.To facilitate the comparison, the units of precipitation rate and temperature tendencies were converted to mm·d$^{-1}$ and K·d$^{-1}$ from mm·s$^{-1}$ and K·s$^{-1}$, respectively, by applying a conversion factor of 86,400 (24 × 3600). Similarly, the water vapor mixing ratio (rqvcuten), cloud water mixing ratio (rqccuten), and rain water mixing ratio (rqrcuten) due to convection were multiplied by 86,400,000 (24 × 3600 × 1000) to convert from kg·kg$^{-1}$·s$^{-1}$ to g·kg$^{-1}$·d$^{-1}$. Among the output variables listed in Table 1, the variable w0avg, is excluded as it is calculated using an equation with the ground truth as input in this offline validation. Hence, evaluating w0avg in the offline evaluation is unnecessary.

Among all variables illustrated in Figure 4, the variable "nca" exhibits a significantly higher RMSE of 4.32, with data points widely dispersed across a wide range of values. This suggests that accurately predicting convection poses a considerable challenge. To eliminate the dependency on time steps, "nca" is divided by the model's time step of 15 seconds before proceeding with plotting and statistical evaluations. The precipitation rate demonstrates the highest correlation coefficient and minimal variability, as most data points cluster closely around the 1:1 line. While temperature and the four moisture tendencies exhibit some degree of variability, the majority of data points align closely to the 1:1 line. The correlation coefficient of convection trigger is 0.91, not shown in Figure 4. Overall, the ML-based MSKF scheme shows a strong correlation with the original MSKF scheme for all examined variables, with correlation coefficients consistently higher than 0.91. This indicates that the the ML-based MSKF scheme has the potential to replace the original scheme.

To obtain a comprehensive understanding of the vertical distribution of errors, Figure 5 presents the vertical profiles of error statistics associated with convective tendencies. The solid and dotted lines in the figure represent the MAE and MBE of tendencies at each vertical layer, respectively. Additionally, the shaded area corresponds to the 5th and 95th percentiles of differences between tendencies predicted by the ML-based MSKF predicted scheme and the original MSKF scheme, respectively. The distribution of vertical errors in all tendencies exhibits a notable uniformity, with higher variance observed within the pressure layers between 800 and 1,000 hPa. These pressure layers correspond to the atmospheric layer where convection occurs most frequently. Due to the significantly lower cloud and rain content compared to water vapor in the atmosphere, the error magnitudes for rqccuten and rqrcuten are considerably lower than those observed for rqvcuten.

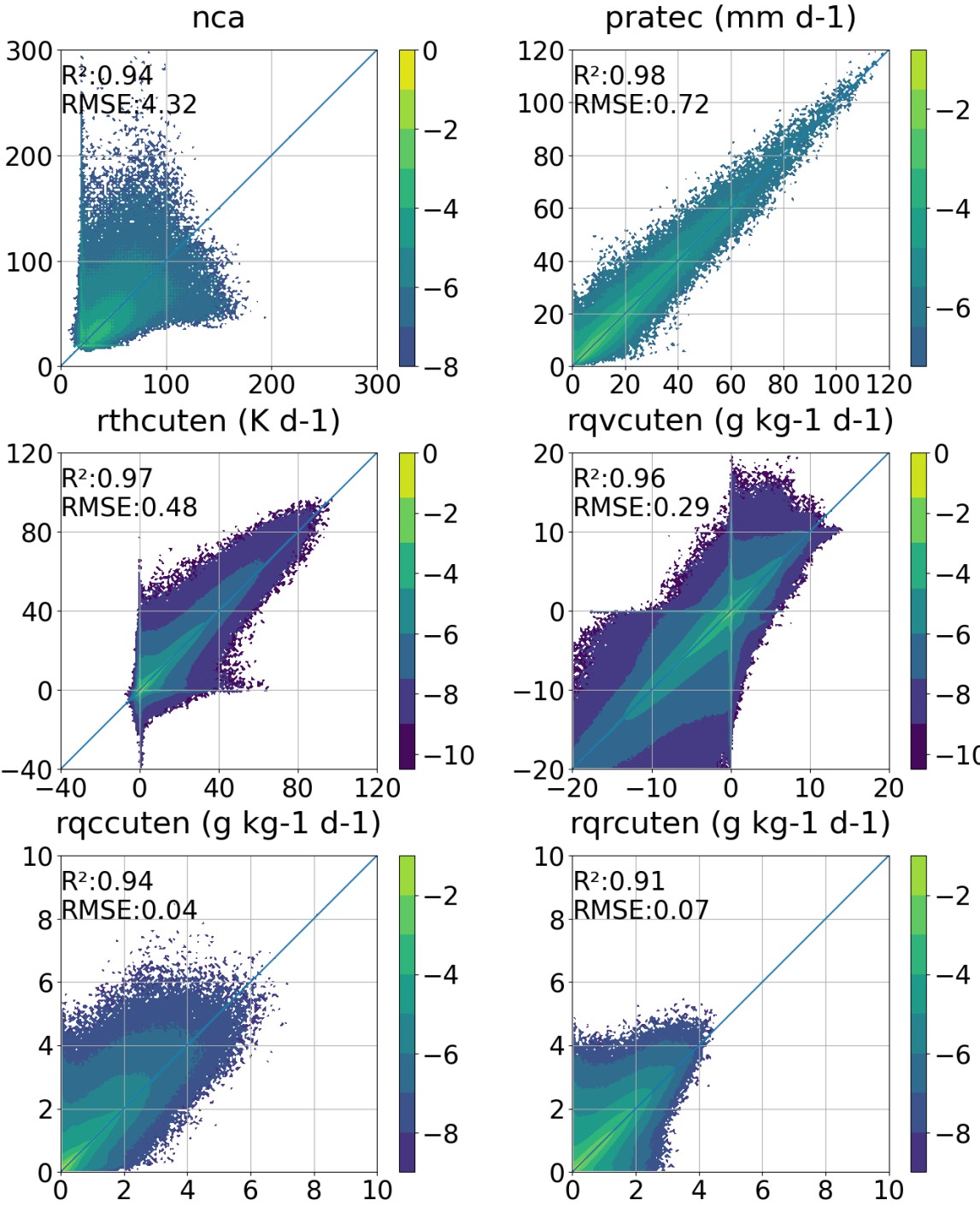

**Figure 4.** Comparison of the predicted (y axis) and true (x axis) nca, pratec, rthcuten (first column), rqvcuten (second column), rqccuten (third column), and rqrcuten for using validation data in the offline setting. Colors indicate the proportion of samples across the entire testing dataset, with values on the colorbar normalized through the application of a logarithm base 10.

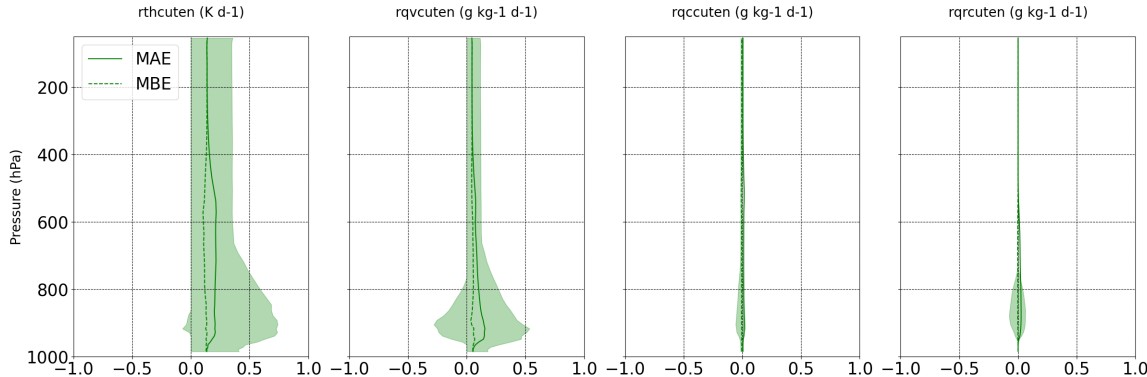

**Figure 5.** Vertical profiles of the statistics in rthcuten (first column), rqvcuten (second column), rqccuten (third column), and rqrcuten (fourth column) using validation data in the offline setting data using ML-based emulators. The solid and dotted lines show the MAE and MBE profile, respectively, and the shaded area indicates the 5th and 95th percentile of differences (prediction—target) at each layer.

## 4.2 Prognostic validation

This subsection presents the performance of the ML-based MSKF scheme in the online setting.

The ML-based MSKF scheme was integrated into the WRF model as a substitute for the original MSKF scheme to simulate convective processes. Utilizing the WRF-ML coupler (Zhong et al., 2023a), this novel ML-based MSKF scheme was seamlessly incorporated into the WRF framework. A comparative analysis was conducted by initializing both the modified WRF model, which incorporates the ML-based scheme, and the original WRF model on June 12, 14, 16, and 18, 2022, for simulations extending over 168 hours. It is worth mentioning that these simulations were performed independently of the training dataset, ensuring the evaluation of the scheme's generalization capability.

Figure 6 presents the averaged spatial forecasts for predictions generated by the original WRF model. These forecast results include the accumulations of both convective precipitation ($RAINC$) and non-convective precipitation ($RAINNC$) over a 12-hour period, along with the 2-meter temperature ($T2M$) at 24, 72, 120, and 168 hours. The figure also demonstrates the mean absolute difference (MAD) between WRF simulations coupled with the ML-based MSKF scheme and those utilizing the original MSKF scheme. Within the spatial forecasts, red and blue patterns signify the magnitudes of the forecasted values, whereas in the spatial differences, these colors denote the positive and negative biases in the ML-based simulations, respectively. Green patterns suggest minimal deviation from the original WRF simulations. Furthermore, we calculate a domain-averaged MAD to evaluate the overall performance of the ML-based scheme in prognostic simulations. Generally, the differences are small, indicating good agreement between WRF simulations coupled with the ML-based MSKF scheme and the original WRF simulations. Notably, the differences do not increase with the progression of simulation time, as evidenced by a comparable domain-averaged MAD at 168 forecast hours compared to that at 24 forecast hours. These findings suggest that the ML-based MSKF scheme achieves stable prognostic simulations.

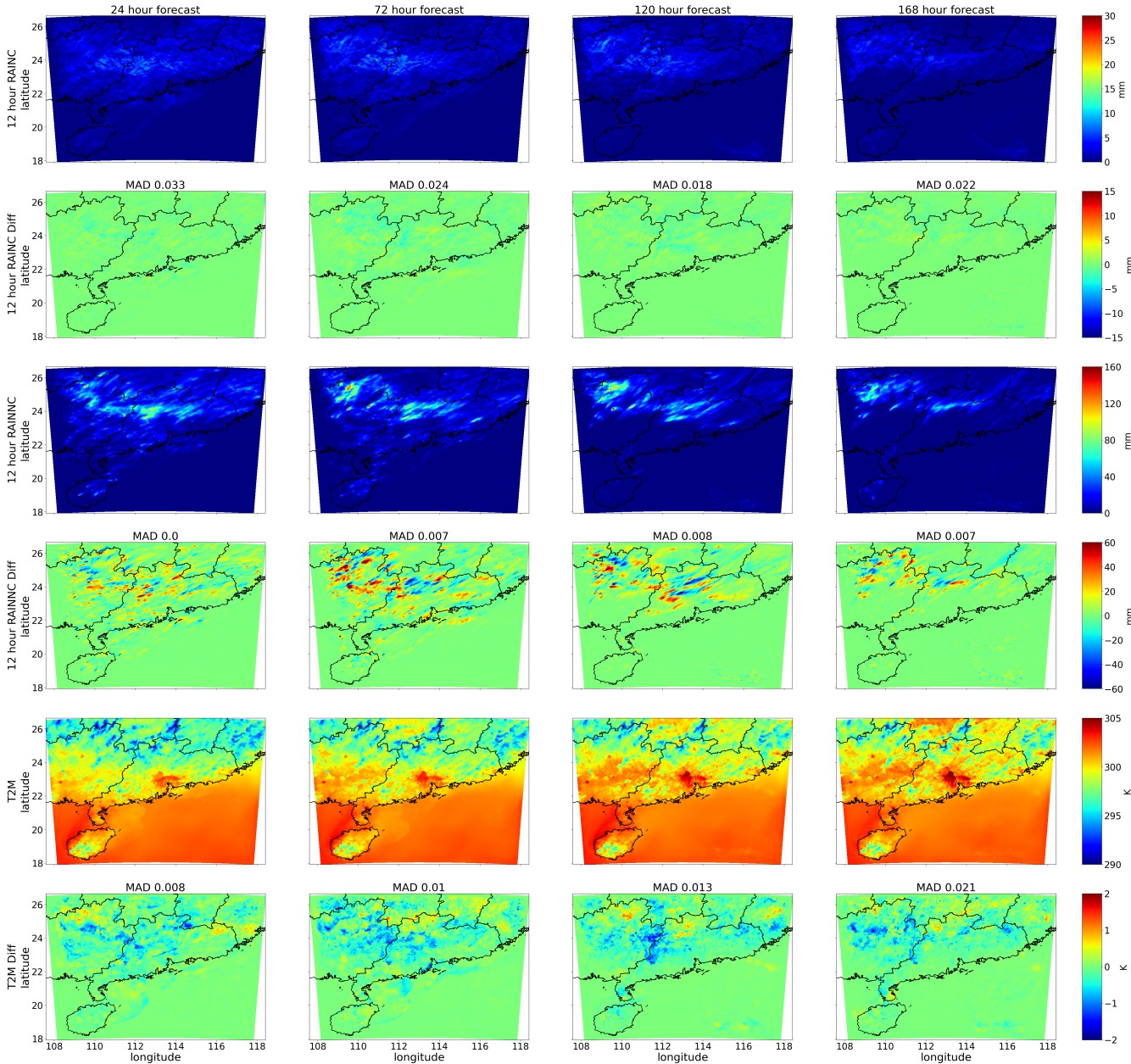

**Figure 6.** Spatial map of the average WRF simulations using the original MSKF scheme (in the first, third, and fifth rows) along with the average MAD between WRF simulations coupled with the ML-based MSKF scheme and WRF simulation with the original MSKF scheme (in the second, fourth and sixth rows). The simulations are shown for the 12-hour accumulated convective precipitation ($RAINC$) in the first and second rows, the 12-hour accumulated non-convective precipitation ($RAINNC$) in the third and fourth rows, and the 2-meter temperature ($T2M$) at forecast lead times of 24 hours (first column), 72 hours (second column), 120 hours (third column), and 168 hours (fourth column).

Figure 7 provides a comparative analysis of domain-averaged time series forecasts from both the original WRF simulations and WRF simulations coupled with the ML-based MSKF scheme. This comparison includes 6-hour accumulations of $RAINC$ and $RAINNC$, as well as $T2M$ forecasts. The results demonstrate that WRF simulations coupled with the ML-based MSKF schemes are in close alignment with the original WRF simulations, particularly in capturing the diurnal variations of $RAINC$, $RAINNC$, and $T2M$. Notably, the $T2M$ forecasts demonstrate remarkable consistency, underscoring the efficacy of the ML-based MSKF scheme in maintaining the predictive accuracy of the original scheme.

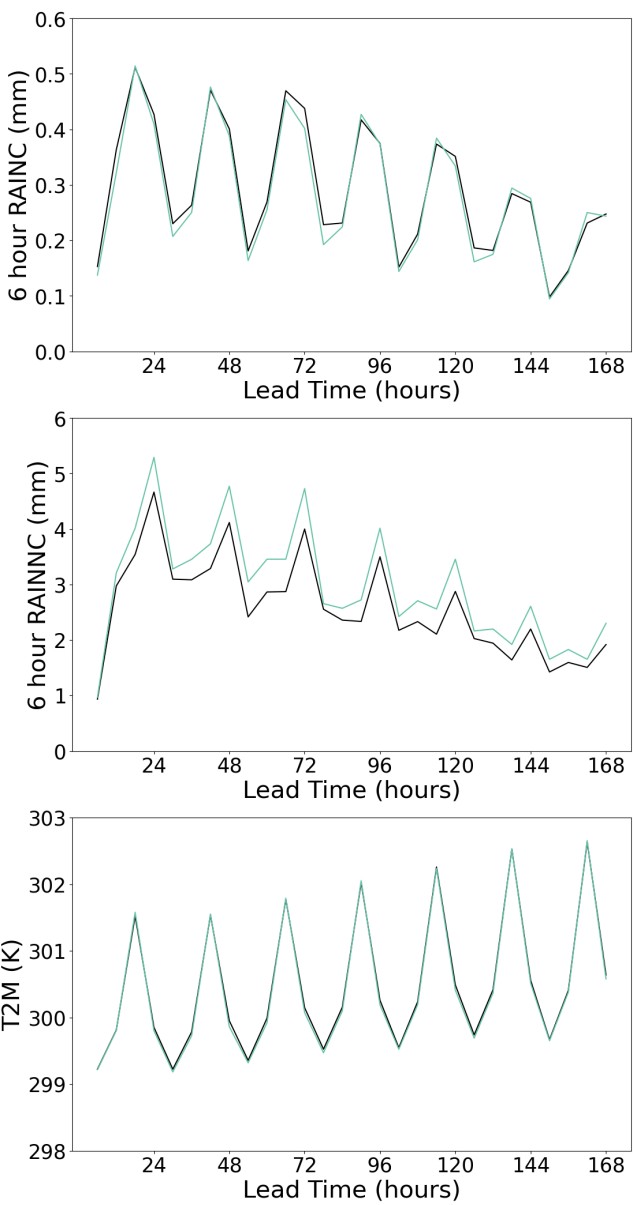

**Figure 7.** Comparison of domain-averaged forecasts derived from the original WRF simulations (black lines) and WRF simulations coupled with the ML-based MSKF scheme (light green lines) of 6-hour accumulated $RAINC$ (first row) and $RAINNC$ (second row), along with $T2M$ (third row).

## 5 Conclusions

In this paper, we proposed a multi-output Bi-LSTM model to develop a ML-based MSKF scheme for predicting convection
trigger and reproducing the convective process in the gray zone. The model is trained on data generated by the WRF simulations
at a spatial resolution of 5 km, covering the South China region. The output variables of the ML-based MSKF scheme are
identical to those of the original MSKF scheme, encompassing cloud relaxation time ("nca"), precipitation rate ("pratec"),
time-step convective precipitation ("raincv"), and convective tendencies. This ML-based scheme ensures physical consistency
among all output variables by incorporating a post-processing module to refine the output from the Bi-LSTM model. Offline
validation demonstrates the excellent performance of the ML-based MSKF scheme. Furthermore, the ML-based MSKF scheme
is coupled with the WRF model using WRF-ML coupler. The WRF simulations coupled with the ML-based MSKF scheme is
compared against the WRF simulation with the original MSKF scheme. Results shows that the ML-based scheme can generate
forecasts similar to the original ML scheme in online settings, showing the potential substitution of the MSKF scheme by ML
models in gray-zone.

This study demonstrates the feasibility of employing ML models as substitutes for conventional CP scheme within the high-
resolution weather forecasting model. Future efforts will focus on the development of ML models, based on data generated
by super-parameterization or cloud-resolving models, to replace conventional CP schemes in weather forecasting models (see
Appendix B). The objective of this substitution is to reduce uncertainties and improve performance of weather forecast models.

*Code and data availability.* The source code for the WRF model version 4.3 used in this study is available at https://doi.org/10.5281/zenodo.
10039053 (Skamarock et al., 2023). The source code and data used in this are available at https://doi.org/10.5281/zenodo.10032404 (Zhong
et al., 2023b).

## Appendix A: Comparison against classification only and regression only models

Two separate Bi-LSTM models were trained with slight modifications to the multi-output Bi-LSTM model illustrated in Figure
3. The first model aimed to predict convection triggers alone, termed Bi-LSTM-trigger, while the second model aimed to
predict convective tendencies, termed Bi-LSTM-tendency. In predicting convection trigger, both the Bi-LSTM-trigger model
and the multi-output Bi-LSTM model demonstrated comparable accuracy, as observed in Figures A1 and A2. However, while
the convection trigger predicted by the Bi-LSTM-trigger model were indistinguishable from those of the multi-output Bi-
LSTM model, the former failed to accurately predict corresponding convective tendencies. Consequently, it cannot serve as a
replacement for convection schemes within NWP models.

Figures A3 and A4 present snapshots of rthcuten and rqvcuten predicted by the Bi-LSTM-tendency model. These figures re-
veal that the Bi-LSTM-tendency model predicts non-zero values across nearly the entire domain. Since the Bi-LSTM-tendency
model exclusively focuses on predicting convective tendencies, convection trigger are derived using certain threshold values.
The spatial distribution of these triggers is notably influenced by the choice of threshold values, and the patterns of convec-

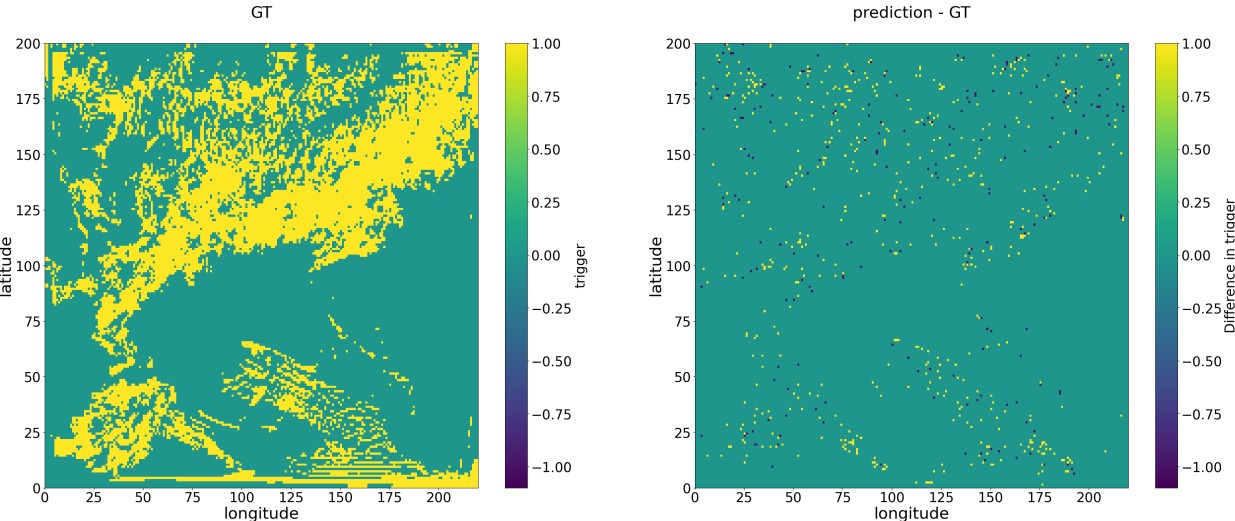

**Figure A1.** Snapshot example of convection trigger, with the left column showing the ground truth (GT), and the right column showing the difference between convection trigger as predicted by the Bi-LSTM-trigger model and ground truth values, for the 25-hour WRF simulation initialized at 12UTC on May 20th, 2021.

tion trigger derived from rthcuten and rqvcuten exhibit considerable discrepancies. This confirms that models based solely on regression yield inconsistent tendencies. In contrast, the multi-output Bi-LSTM model does not encounter the aforementioned issues of the Bi-LSTM-tendency model and generates a more consistent spatial pattern of rthcuten and rqvcuten (see Figures A5 and A6).

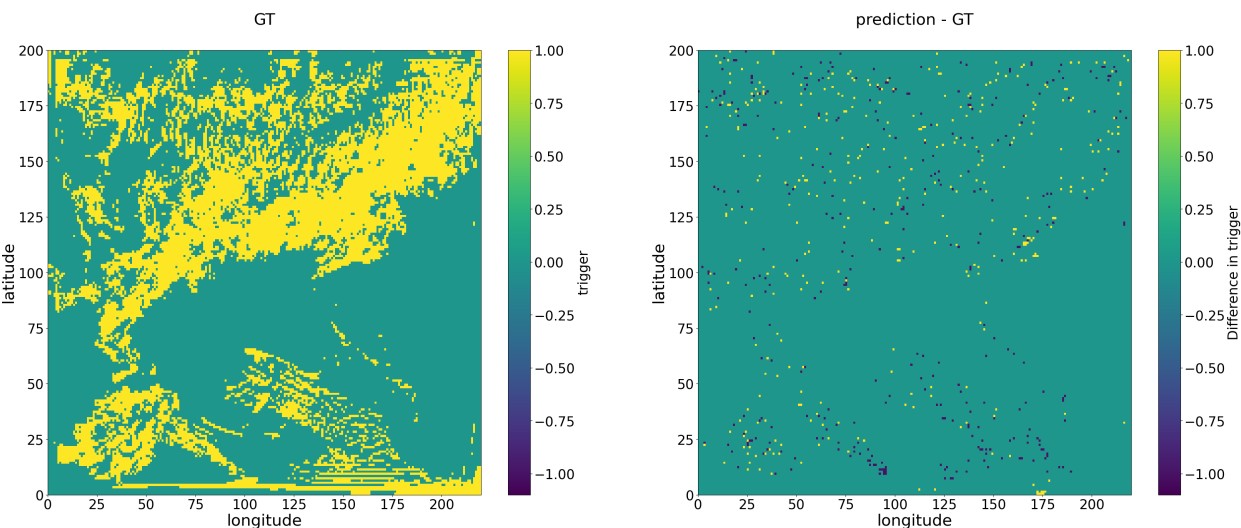

**Figure A2.** Snapshot example of convection trigger, with the left column showing the ground truth (GT), and the right column showing the difference between convection trigger as predicted by the multi-output Bi-LSTM model and ground truth values, for the 25-hour WRF simulation initialized at 12UTC on May 20th, 2021.

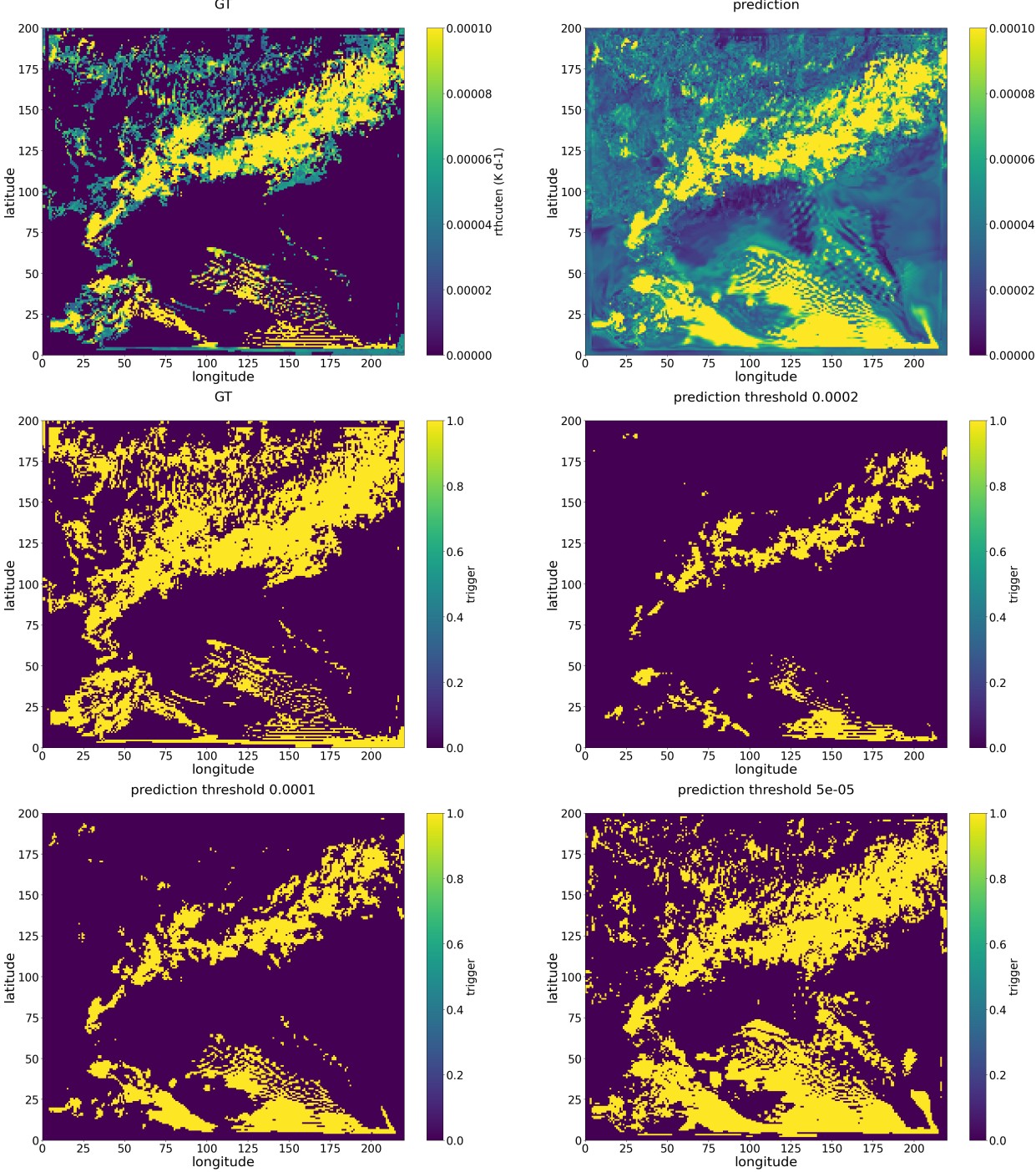

**Figure A3.** Snapshot examples of rthcuten summed along the vertical direction, with the top left panel showing the GT values and the top right panel showing the rthcuten predicted by the Bi-LSTM-tendency model, for the 25-hour WRF simulation initialized at 12UTC on May 20th, 2021. Similarly, snapshot examples of trigger, with the GT shown in the middle left panel, and the predictions from the Bi-LSTM-tendency model using varying threshold values of rthcuten shown in the middle right column, bottom left panel, and bottom right panel, respectively.

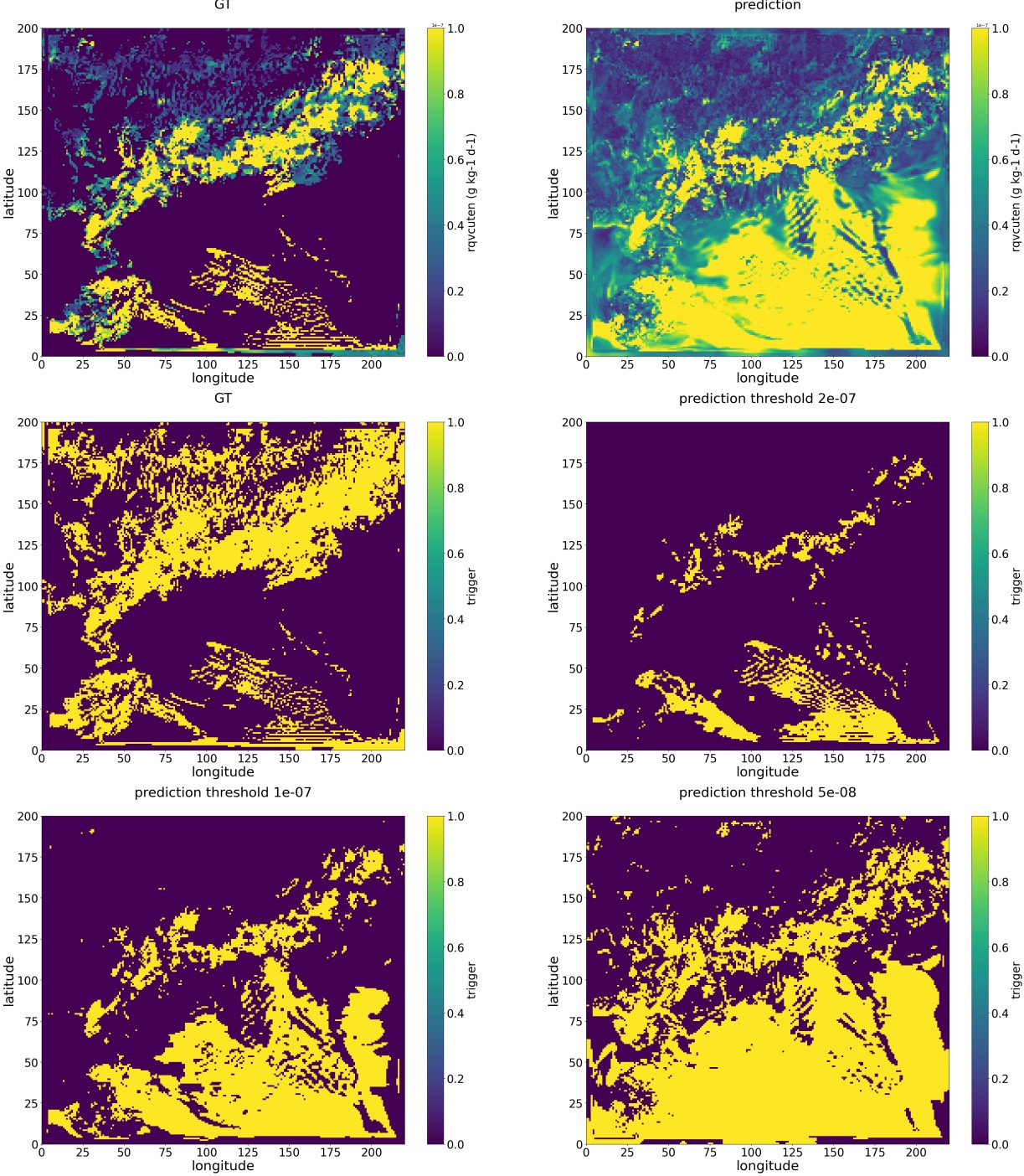

**Figure A4.** Snapshot examples of rqvcuten summed along the vertical direction, with the top left panel showing the GT values and the top right panel showing the rqvcuten predicted by the Bi-LSTM-tendency model, for the 25-hour WRF simulation initialized at 12UTC on May 20th, 2021. Similarly, snapshot examples of trigger, with the GT shown in the middle left panel, and the predictions from the Bi-LSTM-tendency model using varying threshold values of rqvcuten shown in the middle right column, bottom left panel, and bottom right panel, respectively.

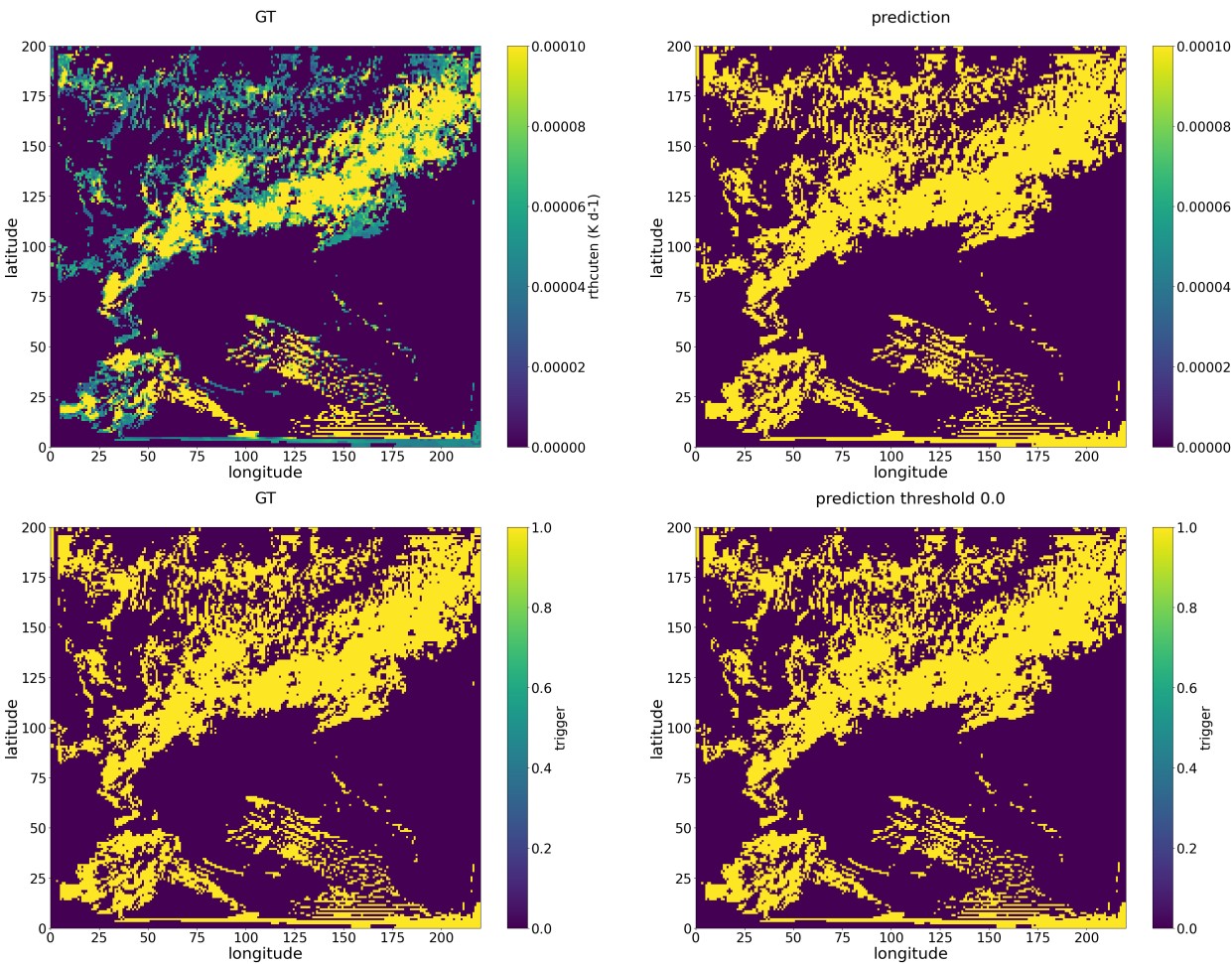

**Figure A5.** Snapshot examples of rthcuten summed along the vertical direction, with the top left panel showing the GT values and the top right panel showing the rthcuten predicted by the multi-output Bi-LSTM model, for the 25-hour WRF simulation initialized at 12UTC on May 20th, 2021. Similarly, snapshot examples of trigger, with the GT shown in the bottom left panel, and the predictions from the multi-output Bi-LSTM model using a threshold value of 0 shown in the bottom right panel.

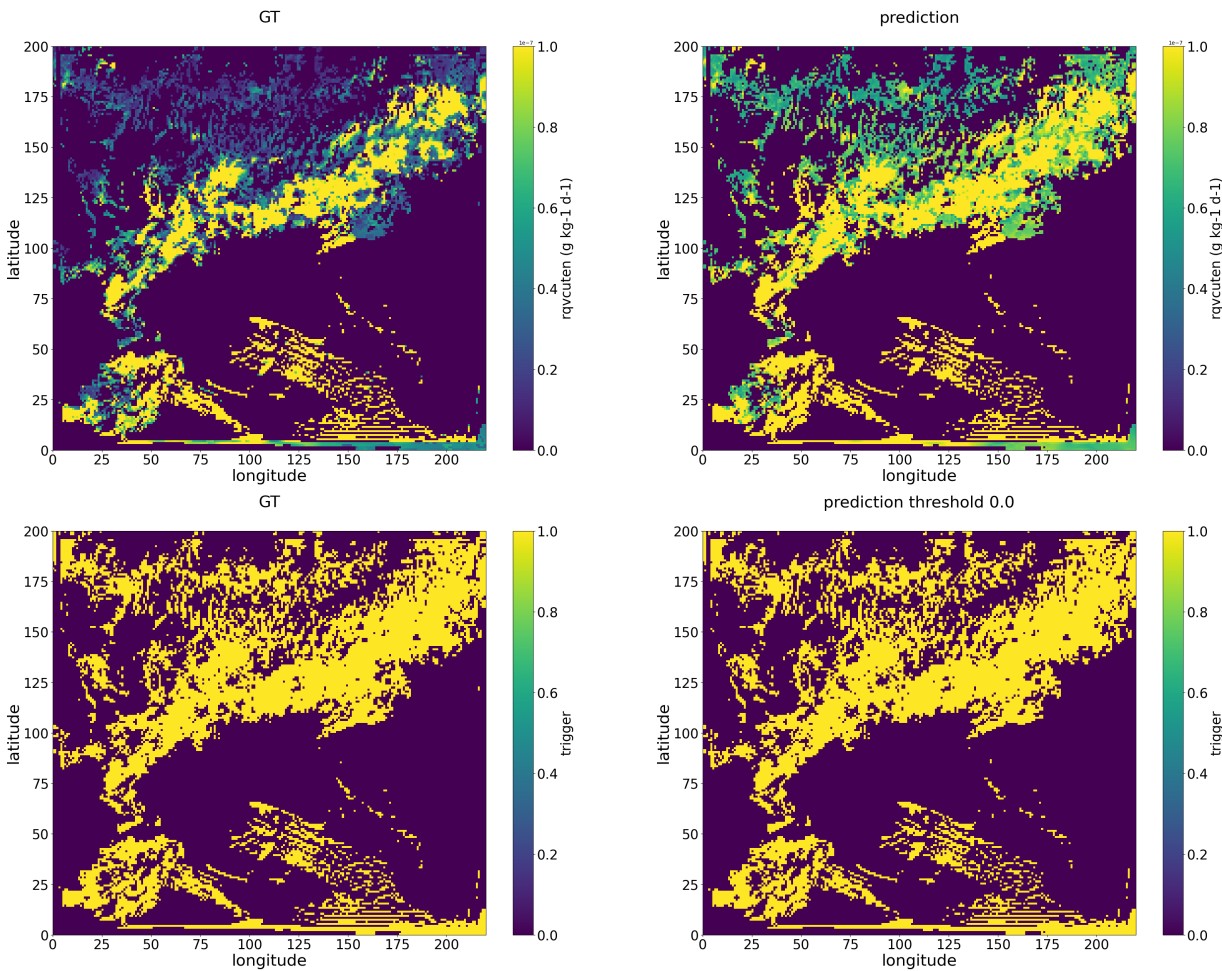

**Figure A6.** Snapshot examples of rqvcuten summed along the vertical direction, with the top left panel showing the GT values and the top right panel showing the rqvcuten predicted by the multi-output Bi-LSTM model, for the 25-hour WRF simulation initialized at 12UTC on May 20th, 2021. Similarly, snapshot examples of trigger, with the GT shown in the bottom left panel, and the predictions from the multi-output Bi-LSTM model using a threshold value of 0 shown in the bottom right panel.

## Appendix B: Significance of using ML-based parameterization to replace super-parameterization and cloud-resolving model

Khairoutdinov et al. (2009) employed Large-Eddy Simulation (LES) to model deep tropical convection over an area of approximately 205 km × 205 km, focusing particularly on maritime regions. They conducted a benchmark simulation spanning 24 hours, with a spatial resolution of 100m and 256 vertical levels. This benchmark simulation utilized 2048 processors and took approximately 6 days of wall-clock time to complete. Additionally, we attempted a cloud-resolving simulation, covering a domain of 600 km × 500 km domain with a grid spacing of 500 m (resulting in a grid of 1200 x 1000 points) and employing

45 vertical levels. The wall-clock time for this simulation was approximately 40 times the forecast time (dt = 2 seconds). For a single 36-hour simulation, the computational time is around 60 days, which far exceeds our current computational resources. Therefore, implementing machine learning-based parameterization would offer a significant advantage in reducing computational costs when replacing the super-parameterization scheme or cloud-resolving model.

*Author contributions.* X.Z. trained the deep learning models and calculate the statistics of model performance. X.Y. and X.Z. conducted the
355 WRF simulations to provide dataset for training and evaluation, and offered valuable suggestions on the model training and paper revision. X.Z. and X.Y. wrote, reviewed and edited the original draft; X.Z., X.Y., and H.L. supervised and supported this research and gave important opinions. All of the authors have contributed to and agreed to the published version of the manuscript.

*Competing interests.* The authors declare no conflict of interest.

*Acknowledgements.* This work was supported by Basic and Applied Basic Research Foundation of Guangdong Province, under Grant No.
2021A1515012582. We are thankful for Mesoscale and Microscale Meteorology Laboratory (MMM) at NCAR for developing and sharing the WRF source codes.

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
