# Peer review of "Machine Learning Parameterization of the Multi-scale Kain-Fritsch (MSKF) Convection Scheme and stable simulation coupled in WRF using WRF-ML v1.0"

_EGUsphere, 2023_

## Author Comment (AC1)

**Reply to reviewers' comments**

We thank the reviewer for the time spent on reviewing this manuscript and for providing helpful comments and suggestions.

**Reviewer #1**

This manuscript develops and evaluates a ML-based surrogate for the MSKF convection scheme in WRF. The text is well written and the approach seems novel. I am not an expert on ML, so my comments are all on the atmospheric modeling/parameterization side of the work. My major comments are as follows,

1. Why is the goal to emulate what a conventional convection parameterization does in the first place? I ask because the ML scheme presented in the paper seems designed and trained to emulate MSKF performance at 5-km resolution. Why not just run MSKF directly? What's the need to run this ML-based emulator? Is it cheaper? Is it better? Please elaborate a little more on that.

   **Response: Conventional convective parameterization (CP) schemes, including MSKF scheme, are founded on numerous assumptions and are best with considerable uncertainties. The ultimate goal of this research is to develop a ML based parameterization scheme, trained on dataset derived either from a super-parameterization or a cloud-resolving model. This study represents a preliminary exploration into the feasibility of utilizing a ML model as a substitute for conventional CP schemes in weather forecasting models. It evaluates the model's performance, both offline and online, in comparison to the conventional CP schemes. Additionally, we also investigate the stability of the WRF simulation when coupled with such a ML model. This is achieved by conducting simulations over extended period, the results of which are shown below.**

2. Is the performance of MSKF good for the area and cases the authors are interested in? If so the authors should provide good evidence for it.

   **Response: The performance of MSKF is good for high-resolution forecasts at spatial resolution similar to what we used in this paper. We provided evidence as follows:**

   **Zheng et al. (2016) demonstrated that the MSKF scheme, referred to as the updated KF scheme in their paper, yielded superior forecast performance in high-resolution forecasts, specifically regarding the location and intensity of precipitation at spatial resolution of 3 and 9 km.**

[Figure]

Furthermore, Ou et al. (2020) ran WRF simulations using various CP schemes and a control simulation without any CP schemes. The CP schemes evaluated include the Grell-3D Ensemble (Grell), New Simplified Arakawa-Schubert (NSAS), and MSKF. These simulations underwent comparative analysis against both in-situ observations and satellite products. Ou et al. (2020) illustrated in their Figure 11 that the MSKF scheme, noted for its scale-awareness, outperforms other CP schemes in simulating precipitation, achieving the lowest RMSE values. This superiority was particularly evident in terms of mean intensity and diurnal cycles of precipitation. Furthermore, the spatial distribution of peak precipitation timing across all the experiments, most notably in the MSKF experiment, showed enhanced agreement with satellite observations.

[Figure]

Ou, T., D. Chen, X. Chen, C. Lin, K. Yang, H. W. Lai, and F. Zhang, 2020: Simulation of summer precipitation diurnal cycles over the Tibetan Plateau at the gray-zone grid spacing for cumulus parameterization. *Clim. Dyn.*, **54**, 3525–3539, https://doi.org/10.1007/s00382-020-05181-x.

Zheng, Y., K. Alapaty, J. A. Herwehe, A. D. Del Genio, and D. Niyogi, 2016: Improving high-resolution weather forecasts using the Weather Research and Forecasting (WRF) model with an updated Kain-Fritsch scheme. *Mon. Weather Rev.*, **144**, 833–860, https://doi.org/10.1175/MWR-D-15-0005.1.

**Also, we added the following sentences in the Introduction:**
**"To enhance prediction accuracy in the gray zone, researchers have developed scale-aware CP schemes. These schemes dynamically parameterize convective processes based on the horizontal grid spacing, thus facilitating seamless transitions between different spatial scales. A pivotal study by Jeworrek et al. (2019) demonstrated that two specific scale-aware CP schemes, Grell-Freitas (Grell and Freitas, 2014) and multi-scale Kain-Fritsch (MSKF) (Zheng et al., 2016), surpassed conventional CP schemes in predicting both the timing and intensity of precipitation over the Southern Great Plains of the United States. Additionally, Ou et al. (2020) showed that the MSKF scheme outperformed other CP schemes, including Grell-3D Ensemble (Grell and Dévényi, 2002) and New Simplified Arakawa-Schubert (Han and Pan, 2011), in precipitation simulation. This was evidenced by its lower root mean squared error (RMSE) values when compared against in-situ observations and satellite data. Despite the increasing adoption of these scale-aware schemes due to their superior performance, it is crucial to acknowledge that their efficacy also rely on various empirical parameters (Villalba-Pradas and Tapiador, 2022). Therefore, developing specialized CP schemes for the gray zone in NWP models continues to be a significant challenge."**

**Minor comments (mainly on the introduction part, which is otherwise quite well-written) follow,**

1. Line 37: "Nevertheless ... These conflicting findings typically...", Did the Schwartz paper also use CP in their simulations or not? Either way I don't quite see the conflicting part here ...

   **Response: Our objective is to convey that although some studies suggest no noticeable benefit in employing finer grid spacing, others demonstrate enhanced forecast accuracy with increased horizontal resolution. Consequently, we have revised the previously ambiguous paragraph to enhance clarity as follows:**

   **"There is ongoing debate regarding the efficacy of employing convection parameterization (CP) within the gray zone. Several studies (Chan et al., 2013;**

Johnson et al., 2013) have found that reducing horizontal grid spacing to less than 4 km while using CP scheme, does not enhance and may even degrade, precipitation forecast performance. In contrast, other studies (Lean et al., 2008; Roberts and Lean, 2008; Clark et al., 2012) showed that forecasts with a horizontal grid spacing of 1 km yielded more accurate spatial representation of accumulated rainfall over 48 hours when compared to 12 km and 4 km grid spacing. This discrepancy in research findings, with some indicating no benefit from finer grid spacing and others suggesting improved forecast accuracy, seems to stem from the application of the CP at scales beyond its originally intended operational range."

2. Line 102: "Furthermore, all previous studies have predominantly focused on using CP schemes in GCM models for climate forecasting. Moreover, the choice of CP schemes significantly influences the uncertainty in precipitation forecasts within weather forecasting models. The complexity of the CP schemes also surpasses those applied in climate models (Arakawa, 2004)." I don't think the last statement is generally true. Also the logic doesn't seem to flow among these few sentences.

**Response: According to Arakawa, (2004), formulating cumulus parameterization (CP) schemes in high-resolution models presents greater challenges than in coarse-resolution models. This complexity arises as conventional CP scheme are based, either explicitly or implicitly, on the assumption that the horizontal grid size and the time interval for implementing physics are significantly larger and longer than the size and lifetime of individual moist-convective elements. However, this assumption does not hold true for high resolution model. Consequently, CP schemes for high resolution models must include dependencies on both horizontal resolution and the time interval for implementing physics. To enhance clarity and logic, the previously ambiguous paragraph has been revised as follows:**

**"The primary focus of previous research has been on focused on replacing CP schemes in GCM models with ML models for climate forecasting. However, the complexity of CP schemes in weather forecasting models is considerably greater than those in GCMs (Arakawa, 2004). Generally, CP schemes in GCMs, whether in explicit or implicit form, assume that both the horizontal grid size and the temporal intervals for physics implementation are significantly larger and longer compared to the grid size and duration of individual moist-convective elements. In contrast, CP schemes in high-resolution models must account for dependencies on both the model's resolution and the time interval for implementing the physics (Arakawa, 2004). The ultimate objective is to develop ML models, based on data from super-parameterization or cloud-resolving models, to replace conventional CP schemes in weather forecasting models. This replacement seeks to reduce uncertainties and improve the efficacy of ML parameterizations. This study**

represents an initial effort to use a ML model as an alternative to conventional CP schemes in weather forecasting models."

**Reviewer #2**

This study provides an ML multiscale Kain-Fritsch convection parameterization and applies it into WRF. The results indicate that the ML parameterization yields comparable outcomes to the original WRF simulations. However, the motivation behind this study is unclear, and the evaluation of the ML parameterization lacks thoroughness. As a result, I recommend a major revision to address these issues.

**Major comments:**

3. This study focuses on the development of a new machine learning (ML) convection parameterization. However, the training dataset used in this study is derived from the WRF simulation, which may not provide significant benefits for the ML parameterization. The dataset essentially serves as a surrogate for the old parameterization. Instead, a more suitable choice for the dataset could be from a super-parameterization or cloud resolve model, as it could help reduce uncertainties and enhance the ML parameterization's performance.

   **Response: We agree with the reviewer's comment that the ML parameterization, trained on a dataset derived from the WRF simulation, essentially serves as a surrogate for traditional parameterization. Our ultimate objective is to have a ML parameterization scheme trained on dataset from either a super-parameterization or a cloud-resolving model. This present study represents an initial attempt into the feasibility of employing a ML model as an alternative to conventional CP schemes within weather forecasting models. Additionally, it examines the stability of the WRF simulation coupled with such a ML model over extended period. To elucidate this point further, we have expanded the Introduction and Conclusion sections with additional sentences:**

   **"The ultimate objective is to develop ML models, based on data from super-parameterization or cloud-resolving models, to replace conventional CP schemes in weather forecasting models."**

   **"This study demonstrates the feasibility of employing ML models as alternatives for conventional CP scheme within the high-resolution weather forecasting model. Future efforts will focus on the development of ML models, based on data generated by super-parameterization or cloud-resolving models, to replace conventional CP schemes in weather forecasting models. The objective of this substitution is to reduce uncertainties and improve performance of weather forecast models."**

4. It would be beneficial to conduct a comprehensive evaluation of the ML convection parameterization, considering factors beyond just the mean state. In addition to assessing the accuracy of tendencies, it is important to evaluate the parameterization's ability to accurately represent triggered convection. Furthermore, examining the diurnal cycle of precipitation and interpretability of the ML parameterization can provide valuable insights. Additionally, when coupling machine learning parameterization with dynamics, it is crucial to investigate potential issues with simulation stability. Have you tested the parameterization for longer simulations to assess its stability over extended time periods?

**Response: As suggested by the reviewer, we evaluate the parameterization's ability to accurately represent triggered convection by plotting the diurnal variation of precipitation and 2-meter temperature. Additionally, we have incorporated further analysis as below.**

**"Figure 7 provides a comparative analysis of domain-averaged time series forecasts from both the original WRF simulations and WRF simulations coupled with the ML-based MSKF scheme. This comparison includes 6-hour accumulations of RAINC and RAINNC, as well as T2M forecasts. The results demonstrate that WRF simulations coupled with the ML-based MSKF schemes are in close alignment with the original WRF simulations, particularly in capturing the diurnal variations of RAINC, RAINNC, and T2M. Notably, the T2M forecasts demonstrate remarkable consistency, underscoring the efficacy of the ML-based MSKF scheme in maintaining the predictive accuracy of the original scheme."**

[Figure]

**Figure 7.** Comparison of domain-averaged forecasts derived from the original WRF simulations (black lines) and WRF simulations coupled with the ML-based MSKF scheme (light green lines) of 6-hour accumulated $RAINC$ (first row) and $RAINNC$ (second row), along with $T2M$ (third row).

Also, we conducted 7-day simulations to evaluate the stability of the ML based MSK scheme over extended time periods. As a result, we updated the original paper by replacing Figure 6 with the revised figure presented here. The updated figure illustrates that the difference between the original WRF simulations and those coupled with the ML-based MSKF scheme remains consistent over time. Notably, the domain-averaged MAD at 168 forecast hours is comparable to that observed at 24 forecast hours, indicating no significant increase in the difference as the simulation duration extends.

[Figure]

**Figure 6.** Spatial map of the average WRF simulations using the original MSKF scheme (in the first, third, and fifth rows) along with the average MAD between WRF simulations coupled with the ML-based MSKF scheme and WRF simulation with the original MSKF scheme (in the second, fourth and sixth rows). The simulations are shown for the 12-hour accumulated convective precipitation ($RAINC$) in the first and second rows, the 12-hour accumulated non-convective precipitation ($RAINNC$) in the third and fourth rows, and the 2-meter temperature ($T2M$) at forecast lead times of 24 hours (first column), 72 hours (second column), 120 hours (third column), and 168 hours (fourth column).

5. In this study, the machine learning approach performs multi-task learning, simultaneously handling trigger function classification and tendencies regression. It would be valuable to conduct an ablation study, where the classification and regression tasks are separately evaluated and compared with the multi-task learning approach. This analysis can provide insights into the individual contributions and effectiveness of each task, helping to further understand the benefits and limitations of the multi-task learning approach.

**Response: In the conventional CP scheme, convection tendencies are computed at specific grid points where convection is triggered. However, models based only on regression can yield inconsistent tendencies, resulting in conflicting indications for convection triggering at specific grid points. In contrast, models that rely exclusively on classification are also deficient, as they do not generate the necessary tendencies for the CP scheme. Therefore, a model confined to either classification or regression tasks is inadequate for meeting the CP scheme's requirements. To overcome these limitations, we have developed a multi-task learning approach. To elucidate this point further, we have expanded the subsection "ML model structure" as follows:**

**"Predicting whether convection is triggered as well as modeling convective tendencies and precipitation rate are two main tasks in conventional CP schemes. Regression-based models alone may produce inconsistent tendencies, leading to conflicting signals for triggering convection at specific grid points. Similarly, models solely dependent on classification lack the capacity to generate essential tendencies for an effective CP scheme. Therefore, the development of a ML-based CP scheme necessitates the integration of both a binary classification model for the prediction of convection trigger and a regression model for convective tendencies. To address this, we propose a multi-output Bi-LSTM model capable of concurrently conducting regression and classification predictions (Figure 3)."**

6. In Figure 4, is it correct that r^2 is the coefficient of determination? It seems unusual that although the points in (a) are widely scattered, the r^2 value is very high. Also, while the RMSE difference between (a) and (b) is large, their r^2 values are quite close.

**Response: Following the reviewer's suggestions, we have verified that the calculation of correlation coefficient depicted in Figure 4 is correct. Figure 4 utilizes color to represent the proportion of samples across the entire testing dataset, with the colorbar values normalized through the application of a base 10 logarithm. Therefore, despite the wide scatter of data points in Figure 4(a), a significant majority of these points remain close to 1:1 line. The observed Root Mean Square Error (RMSE) discrepancy between panels (a) and (b) can be attributed to the substantial differences in the maximum and minimum values of nca and pratec, where nca exhibits a broader range of values.**

**Minor comments:**

1. Line 145: Could you please explain the rationale for selecting these particular variables?

**Response: As suggested by the reviewer, we added the following sentence to explain the rationale for selecting these particular variables.**

**"Table 1 presents a comprehensive list of the input and output variables used in this study, aligning with those utilized in the original MSKF scheme"**

2. Line 170: could you explain the trigger condition base on lifting condensation level (LCL), convective available potential energy (CAPE), cloud top and base heights, and entrainment rates?

**Response: In order to better explain the trigger condition based on lifting condensation level (LCL), convective available potential energy (CAPE), cloud top and base heights, and entrainment rates. We added the following sentences as below in the subsection "Description of original MSKF module":**

**"In contrast, a "nca" value below this threshold triggers the MSKF scheme to employ a one-dimensional cloud model. This model calculates a set of variables related to cloud characteristics to evaluate the potential of convection triggering. Essential variables include the lifting condensation level (LCL), convective available potential energy (CAPE), cloud top and base heights, and entrainment rates. The LCL is crucial for determining the emergence of potential convective activities, with a lower LCL favoring more intense convection. CAPE quantifies the buoyant energy available to an air parcel for the formation of deep convective clouds upon reaching its Level of Free Convection (LFC) above the LCL, with higher CAPE values signifying a greater potential for intense convection. The cloud base is generally at the LCL, whereas the cloud top is defined at the altitude where buoyancy becomes negligible. Meanwhile, the vertical extent between the cloud base and top affect the cloud's growth and precipitation potential. The MSKF scheme requires surpassing a specific CAPE threshold to trigger convection. Furthermore, it assesses entrainment rates to measure the impact of ambient air on the evolution of convective system."**

3.  Line 45: 'hsave' to 'have'

**Response: As suggested by the reviewer, we changed 'hsave' to 'have'.**

4.  Caption in Figure 1: '5' to '5°'

**Response: As suggested by the reviewer, we changed '5' to '5°' in Caption of Figure 1.**

5.  Line 149: it could be better to add these 4 variables into Table 1.

**Response: As suggested by the reviewer, we added the these 4 variables into Table 1. The updated Table 1 is shown below:**

**Table 1.** Definition of all the input and output variables, and whether they are surface or 3D variables and their corresponding units. There are 44 model layers.

| Type | Variable name | Definition | type | Unit |
|---|---|---|---|---|
| Input | u | meridional wind component | 3D | m/s |
| | v | zonal wind component | 3D | m/s |
| | w | vertical wind component | 3D | m/s |
| | t | temperature | 3D | K |
| | qv | water vapor mixing ratio | 3D | kg/kg |
| | p | pressure | 3D | Pa |
| | th | potential temperature | 3D | K |
| | dz8w | layer thickness | 3D | m |
| | rho | air density | 3D | $kg/m^3$ |
| | pi | Exner function, which is dimensionless pressure and can be defined as: $\left(\frac{p}{p_0}\right)^{R_d/c_p}$ | | |
| | hfx | upward heat flux at surface | surface | $W/m^2$ |
| | ust | u∗ in similarity theory | surface | $W/m^2$ |
| | pblh | planetary boundary layer height | surface | m |
| Derived Input | $p_{diff}$ | pressure difference between adjacent levels | 3D | Pa |
| | $qv_{sat}$ | saturated water vapor mixing ratio | 3D | kg/kg |
| | rh | relative humidity | 3D | - |
| | trigger | boolean trigger indicating convection triggering | surface | - |
| Input and Output | rthcuten | potential temperature tendency due to cumulus parameterization | 3D | K/s |
| | rqvcuten | water vapor mixing ratio tendency due to cumulus parameterization | 3D | kg/kg/s |
| | rqccuten | cloud water mixing ratio tendency due to cumulus parameterization | 3D | kg/kg/s |
| | rqrcuten | rain water mixing ratio tendency due to cumulus parameterization | 3D | kg/kg/s |
| | rqicuten | cloud ice mixing ratio tendency due to cumulus parameterization | 3D | kg/kg/s |
| | rqscuten | snow mixing ratio tendency due to cumulus parameterization | 3D | kg/kg/s |
| | w0avg | average vertical velocity | 3D | m/s |
| | nca | counter of the cloud relaxation time | 3D | s |
| | pratec | precipitation rate due to cumulus parameterization | surface | mm/s |
| Output | raincv | precipitation due to cumulus paramterization | surface | mm |

6.  Line 182: remove 'data'

**Response: As suggested by the reviewer, we removed the redundant word 'data'**

---

## Referee Report (RR1)

Thank you to the authors for thoroughly addressing the comments. However, further emphasis on the study's contribution to the community would strengthen the manuscript.

1. The authors acknowledge the comment regarding the ultimate goal of developing an ML parameterization trained from superparameterization or cloud-resolved models. An ML parameterization offers two potential advantages: first, it can help reduce uncertainty. Second, it can reduce the computational cost compared to running superparameterization or high-resolution models directly. However, this study presents an ML model as a surrogate for an existing microphysics scheme rather than training it on high-fidelity data, which cannot reduce uncertainty. In order to highlight the contribution, the authors could present the computational efficiency. To better evaluate the ability to reduce costs, the authors could estimate the computational requirements of superparameterization or high-resolution runs to directly compare against the ML model performance. This would strengthen the case for ML as a lower-cost alternative to traditional parameterization approaches.

2. This study presents a novel multi-task ML approach for both trigger function classification and tendency regression. This multi-faceted application of ML could be emphasized as another key contribution of the work. Previous studies have applied ML either to trigger function classification alone (Zhang et al., 2021) or tendencies regression independently (Brenowitz & Bretherton, 2019; Rasp et al., 2018; Wang et al., 2022). As the authors point out that "models based only on regression can yield inconsistent tendencies, resulting in conflicting indications for convection triggering at specific grid points. In contrast, models that rely exclusively on classification are also deficient, as they do not generate the necessary tendencies for the CP scheme". To further validate this assertion, ablation experiments removing each individual task (i.e. classification-only vs regression-only models) could demonstrate the benefits of the proposed multi-task framework. Such an analysis would help substantiate the value of the multi-faceted ML approach over single-task baselines, strengthening the novel aspects highlighted in this work.

Ref:
Brenowitz, N. D., & Bretherton, C. S. (2019). Spatially extended tests of a neural network

parametrization trained by coarse-graining. Journal of Advances in Modeling Earth

Systems, 11(8), 2728–2744. https://doi.org/10.1029/2019ms001711

Rasp, S., Pritchard, M. S., & Gentine, P. (2018). Deep learning to represent subgrid processes in

climate models. Proceedings of the National Academy of Sciences, 115(39), 9684–9689.

https://doi.org/10.1073/pnas.1810286115

Wang, X., Han, Y., Xue, W., Yang, G., & Zhang, G. J. (2022). Stable climate simulations using a realistic general circulation model with neural network parameterizations for atmospheric moist physics and radiation processes. Geoscientific Model Development, 15(9), 3923–3940. https://doi.org/10.5194/gmd-15-3923-2022

Zhang, T., Lin, W., Vogelmann, A. M., Zhang, M., Xie, S., Qin, Y., & Golaz, J.-C. (2021). Improving Convection Trigger Functions in Deep Convective Parameterization Schemes Using Machine Learning. Journal of Advances in Modeling Earth Systems, 13(5), e2020MS002365. https://doi.org/10.1029/2020MS002365

---

## Author Response (AR2)

**Reply to reviewers' comments**

We thank the reviewer for the time spent on reviewing this manuscript and for providing helpful comments and suggestions.

**Reviewer #2**

Thank you to the authors for thoroughly addressing the comments. However, further emphasis on the study's contribution to the community would strengthen the manuscript.

1. The authors acknowledge the comment regarding the ultimate goal of developing an ML parameterization trained from superparameterization or cloud-resolved models. An ML parameterization offers two potential advantages: first, it can help reduce uncertainty. Second, it can reduce the computational cost compared to running superparameterization or high-resolution models directly. However, this study presents an ML model as a surrogate for an existing microphysics scheme rather than training it on high-fidelity data, which cannot reduce uncertainty. In order to highlight the contribution, the authors could present the computational efficiency. To better evaluate the ability to reduce costs, the authors could estimate the computational requirements of superparameterization or high-resolution runs to directly compare against the ML model performance. This would strengthen the case for ML as a lower-cost alternative to traditional parameterization approaches.

   **Response: As suggested by the reviewer, we added the following sentence in the appendix.**

   **Khairoutdinov et al. (2009) employed Large-Eddy Simulation (LES) to model deep tropical convection over an area of approximately 205 km x 205 km, focusing particularly on maritime regions. They conducted a benchmark simulation spanning 24 hours, with a spatial resolution of 100m and 256 vertical levels. This benchmark simulation utilized 2048 processors and took approximately 6 days of wall-clock time to complete. Additionally, we attempted a cloud-resolving simulation, covering a domain of 600 km x 500 km domain with a grid spacing of 500 m (resulting in a grid of 1200 x 1000 points) and employing 45 vertical levels. The wall-clock time for this simulation was approximately 40 times the forecast time (dt = 2 seconds). For a single 36-hour simulation, the computational time is around 60 days, which far exceeds our current computational resources. Therefore, implementing machine learning-based parameterization would offer a significant advantage in reducing computational costs when replacing the super-parameterization scheme or cloud-resolving model.**

2. This study presents a novel multi-task ML approach for both trigger function classification and tendency regression. This multi-faceted application of ML could be emphasized as another key contribution of the work. Previous studies have applied ML either to trigger function classification alone (Zhang et al., 2021) or tendencies regression independently (Brenowitz & Bretherton, 2019; Rasp et al., 2018; Wang et al., 2022). As the authors point out that "models based only on regression can yield inconsistent tendencies, resulting in conflicting indications for convection triggering at specific grid points. In contrast, models that rely exclusively on classification are also deficient, as they do not generate the necessary tendencies for the CP scheme". To further validate this assertion, ablation experiments removing each individual task (i.e. classification-only vs regression-only models) could demonstrate the benefits of the proposed multi-task framework. Such an analysis would help substantiate the value of the multifaceted ML approach over single-task baselines, strengthening the novel aspects highlighted in this work.

**Response: As suggested by the author, we added the following sentences to strengthen the novel aspects of multi-output ML model over single-task baselines.**

**First, we added the following sentence in the abstract:**

**"This multi-output Bi-LSTM model is capable of simultaneously predicting the convection trigger while also modeling the associated convective tendencies and precipitation rates with high performance."**

**Secondly, we added the following sentence in the subsection "ML model structure":**

**"Previous studies have applied ML models to address these objectives, with some dedicated solely to the classification task of convection trigger (Zhang et al., 2021a), while others have independently pursued the regression of convective tendencies (Rasp et al., 2018; Brenowitz and Bretherton, 2019; Wang et al., 2022)."**

**More importantly, we followed the review's suggestion and conducted the ablation experiments by removing each individual task (i.e. classification-only vs regression-only models), and demonstrated the benefits of the proposed multi-task framework. We added the following sentences and figures in the appendix.**

[revised manuscript text omitted]